# Metabolomic Analysis Reveals the Effect of Insecticide Chlorpyrifos on Rice Plant Metabolism

**DOI:** 10.3390/metabo12121289

**Published:** 2022-12-19

**Authors:** Qi’er Mu, Mingxia Zhang, Yong Li, Fayun Feng, Xiangyang Yu, Jinfang Nie

**Affiliations:** 1Guangxi Colleges and Universities Key Laboratory of Food Safety and Detection, College of Chemistry and Bioengineering, Guilin University of Technology, Guilin 541004, China; 2Jiangsu Key Laboratory for Food Quality and Safety-State Key Laboratory Cultivation Base, Ministry of Science and Technology, Nanjing 210014, China; 3Institute of Food Safety and Nutrition, Jiangsu Academy of Agricultural Sciences, 50 Zhongling Street, Nanjing 210014, China

**Keywords:** chlorpyrifos, metabolomics, physiological activity, pathway analysis, plant growth

## Abstract

Pesticides as important agricultural inputs play a vital role in protecting crop plants from diseases and pests; however, the effect of pesticides on crop plant physiology and metabolism is still undefined. In this study, the effect of insecticide chlorpyrifos at three doses on rice plant physiology and metabolism was investigated. Our results revealed that chlorpyrifos cause oxidative stress in rice plants and even inhibit plant growth and the synthesis of protein and chlorophyll at high doses. The metabolomic results suggested that chlorpyrifos could affect the metabolic profiling of rice tissues and a total of 119 metabolites with significant changes were found, mainly including organic acids, amino acids, lipids, polyphenols, and flavonoids. Compared to the control, the content of glutamate family amino acids were significantly disturbed by chlorpyrifos, where defense-related proline and glutathione were significantly increased; however, glutamic acid, N-acetyl-glutamic acid and N-methyl-glutamic acid were significantly decreased. Many unsaturated fatty acids, such as linolenic acid and linoleic acid, and their derivatives lysophospholipids and phospholipids, were significantly accumulated in chlorpyrifos groups, which could act as osmolality substances to help rice cells relieve chlorpyrifos stress. Three organic acids, aminobenzoic acid, quinic acid, and phosphoenolpyruvic acid, involved in plant defenses, were significantly accumulated with the fold change ranging from 1.32 to 2.19. In addition, chlorpyrifos at middle- and high-doses caused the downregulation of most flavonoids. Our results not only revealed the effect of insecticide chlorpyrifos on rice metabolism, but also demonstrated the value of metabolomics in elucidating the mechanisms of plant responses to stresses.

## 1. Introduction

Pesticides are important agricultural inputs that are essential for protecting crop plants from pathogenic microorganisms and insect pests [1]. Chlorpyrifos, an organophosphate insecticide, belongs to second class pesticides and possess moderate toxicity according to the WHO classification [2]. Chlorpyrifos could protect crops (such as vegetables, fruit trees, corn, cotton and rice) against a wide variety of agricultural pests [3,4]. The well-known toxicity of chlorpyrifos occurs by inhibiting cholinesterase [5,6]. The wide usage of chlorpyrifos has resulted in ecological environmental problems becoming increasingly serious [2,7]. This insecticide had a half-life ranging from 10 to 120 days in the environment [8]. The residues of chlorpyrifos in the environment could migrate toward non-target plants and animals, increasing the risk to food safety and human health [7]. It has been reported that chlorpyrifos could affect motor function of the human body and lead to the accumulation of toxins in the liver due to its toxicity [2,9]. In addition, chlorpyrifos causes a reduction in bacterial, fungal, and actinomycete populations and inhibits nitrogen mineralization in soil [10,11]. Though the risk of chlorpyrifos to the environment and human health, specific information regarding the effect of chlorpyrifos on the physiology and metabolism of plants is not currently available.

Metabolomics is a high-throughput method to identify and quantify organic compounds with low molecular weights (<1200 Da) in organisms, which could provide critical information for understanding the relationships of metabolites with other factors such as phenotypes and genes [12]. Recently, metabolomics has been applied to investigate the molecular mechanism of pesticide effects on plant metabolism [13]. Huang, et al. reported that nanocopper pesticides could cause an increase in some defense system signals, such as benzoic acid, gallic acid, and *p*-coumaric acid [14]. Zhang, et al. reported that imidacloprid could enhance plant amino acid metabolism but inhibited saccharide catabolism [15]. Liu et al. revealed that the herbicide butachlor primarily affected carbohydrate metabolism pathways, and the bactericide tricyclazole remarkably influenced fatty acid metabolism pathways [16]. However, the dose-effect of pesticides on crop plant growth and metabolism has not been well examined.

In the present study, the effect of insecticide chlorpyrifos at three doses on rice plant metabolism was systematically investigated. Plant physiological activities, including malondialdehyde (MDA), soluble protein, chlorophyll, and peroxidase (POD) and superoxide dismutase (SOD) activities, in rice tissues were analyzed. The metabolic profiling of rice shoots was detected by LC-QTOF/MS and was analyzed by the MS-DIAL software. Metabolites and metabolic pathways with significant changes were identified to explain the mechanism of rice responses to chlorpyrifos. The objective of this study was to examine whether and how the pesticide chlorpyrifos affects rice plant physiology and metabolism.

## 2. Materials and Methods

### 2.1. Chemicals

HPLC-grade acetonitrile was obtained from Merck (Darmstadt, Germany). Analytical grade standard chlorpyrifos (>99%), formic acid, and ammonium acetate were obtained from Sigma-Aldrich (Sydney, Australia). All test kits for analyzing MDA, soluble protein, and POD and SOD activities were obtained from Nanjing Jiancheng Bioengineering Institute Co., Ltd. (Nanjing, China). The stock standard solutions of chlorpyrifos were prepared in acetonitrile and stored in amber glass vials at 4 °C.

### 2.2. Plant Preparation and Treatment

The germinated *Oryza sativa L.* seeds were transferred into seedling pots containing vermiculite and were cultivated using Hoagland nutrient solution (Coolaber, Beijing, China). After cultivation for 30 days, the rice plants (N = 400) were divided into four groups, which were sprayed with 50 mL pure water containing chlorpyrifos at concentrations of 2.0 mg/L, 5.0 mg/L, and 20 mg/L, and pure water containing 10 μL acetonitrile (as the control group) respectively. After exposure for 48 h, the chlorophyll content was detected using a Plus chlorophyll *SPAD*-*502* meter (Minolta, Osaka, Japan) and 20 measurements for chlorophyll content in each group were counted. Then the shoot tissues were collected for physiological activity analysis (1 g) and metabolomic analysis (3 g). The shoot tissues for metabolomic analysis were quickly frozen via liquid nitrogen, freeze-dried, ground into a powder, and stored at −80 °C. There existed five independent biological replicates for each treatment. In addition, the remaining rice plants (N = 20) from different treatments were continually cultivated for two weeks, and then the wet weight and height of each plant were recorded.

### 2.3. Physiological Activities

The fresh shoot sample (0.2 g) was mixed with the normal saline at a 1:9 *w*/*v* ratio (weight:volume), extracted under low-temperature, and centrifuged for 15 min at 8000 rpm/min and the supernatants were applied to measure physiological activities, MDA, soluble protein, and POD and SOD activities, by test kits according to the manufacturer’s instructions (ref). There exist five independent biological replicates for each treatment.

### 2.4. Metabolomic Analysis

Each sample of freeze-dried shoots (0.1 g) was weighted into a tube containing 4 mL 80% aqueous methanol and was vortexed and shaken for 20 min. The suspension was centrifuged for 10 min at 8000 rpm and was filtered with a 0.22 μm Millipore membrane filter for metabolic profiling analysis.

The metabolic profiling of rice samples was detected by LC-QTOF/MS, Shimadzu LC-20A HPLC coupled with a Sciex TripleTOF 5600+. An electrospray DuoSpray ion source (ESI) was used. A Waters HSS T3 column (2.1 mm × 100 mm, 3.5 µm) was used for chromatographic separations. The mobile phase A was water with 0.1% formic acid and 5 mM ammonium acetate for positive and negative ionization modes, respectively. The mobile phase B was acetonitrile. A following gradient program was performed: 10% B (the first 3.0 min), 10–95% (3.0–21.0 min), and 10% B (28.0–34.0 min). The information-dependent acquisition mode were used to detect the mass spectrometry data for each sample. The mass scans ranged from 60 to 1000 m/z. The source voltage was set to 5500 V and 4500 V for positive and negative ionization modes, respectively. The collision energy (CE) was set to 30 V and −30 V for positive and negative ionization modes, respectively. The volume of each sample injected was 2 μL. The QC sample that prepared by pooling equal amounts of each sample was detected every 5 samples under the same experimental conditions.

All LC-QTOF/MS data files were analyzed using an open source software MS-DIAL. The compound was identified by comparing the similarity of accurate mass number, second-order spectrogram, and retention time in publicly available databases with data-dependent MS/MS acquisition, some of which were further confirmed using standard chemicals. The matrix of MS peak area in positive or negative ionization modes was analyzed using principal component analysis (PCA). The metabolite with significant changes was screened using the PLS-DA model with VIP larger than 1 and *t*-tests with *p* values smaller than 0.05. Metabolic pathways analysis was analyzed using a web-based tool MetaboAnalyst (pathway analysis module) based on the metabolites with significant changes in the chlorpyrifos group compared to the control and referring to the Kyoto Encyclopedia of Genes and Genomes database. Pearson correlation analysis was used for analyzing the relationship among metabolites, plant growth indexes, physiological activities and chlorpyrifos doses, which were computed using the corr() function in MATLAB. Details regarding data acquisition and analysis were performed as described previously [17].

## 3. Results

### 3.1. Plant Growth and Physiological Activities

The physiological activities of rice shoots in the chlorpyrifos treatments at low, middle, and high doses (low, middle, and high, respectively) and the control (CK) are shown in Figure 1. Both POD and SOD were significantly increased in middle and high groups, compared to the control group; while no significant difference between low and control groups was found. The level of MDA in rice shoots was significantly increased with chlorpyrifos doses. The protein content in rice tissues was significantly down-regulated in middle and high groups at 8.06% and 14.7%, respectively, compared to the control group. The chlorophyll content was significantly up-regulated in low and middle groups but significantly down-regulated in the comparison of High/CK. After two weeks of cultivation, chlorpyrifos at low and middle doses had no effect on plant growth and weight but did inhibit plant growth at high doses (Figure 1f–g). These results demonstrated that chlorpyrifos at low doses could slightly affect plant physiology but causes oxidative stress in rice plants at both middle and high doses and inhibited plant growth and the synthesis of protein and chlorophyll at high doses.

### 3.2. Metabolic Profiling Analysis

PCA analysis of the metabolic profiling in positive and negative ion modes is shown in Figure 2a, where the first and second principal components accounted for 32.44% and 14.73% of the total variance, respectively. A clear distinction among metabolic profiling of samples belonging to different groups could be found in PCA score plots. Based on the MS-DIAL, a total of 119 metabolites with significant changes were found, mainly including organic acids, amino acids, lipids, polyphenols, and flavonoids (Figure 2b, Appendix A). Figure 3 displayed the metabolites with significant changes using heatmap analysis. In the comparison of the low and control groups (Low/CK), 56 metabolites were significantly upregulated, and uracil, feruloyl agmatine, 3-aminobenzoic acid, and arginine showed the greatest increase, with a fold change greater than 2.0. In contrast, 17 metabolites were significantly downregulated, and spermidine had the largest decrease at 0.35-fold. When comparing middle and control groups (Middle/CK), 74 metabolites were significantly upregulated, most of which were lipids; 39 metabolites were significantly downregulated, and the top 10 downregulated metabolites were PE (14:0/18:2), N-acetyl-DL-glutamic acid, icariin, N,N-dimethylarginine, PE (18:0/18:2), FA 16:0, methylsuccinic acid, feruloyl agmatine, PC 34:2 and linoleic acid (ranging from 0.105- and 0.693-fold). When comparing high and control groups (High/CK), 49 metabolites were significantly upregulated and primarily included lipids and flavonoids, while FA 20:5 had the largest increase at 4.50-fold, followed by PC (36:3), PE (38:9), PI (17:0/18:2), 3-aminobenzoic acid, FA 22:0, PI (18:1/18:2), PG (18:1/18:2), PI (18:2/18:2) and PG (17:0/18:3); 47 metabolites were significantly downregulated, and the top 10 metabolites were feruloyl agmatine, luteolin-7-O-glucoside, N-methyl-l-glutamate, methylsuccinic acid, biuret, nicotinamide, L-5-oxoproline, niacinamide, cyanidin3-(2G-glucosylrutinoside), and N2,N2-dimethylguanosine (ranging from 0.045 to 0.072-fold).

### 3.3. Metabolic Pathways Analysis

The enriched metabolic pathways in the comparison of Low/CK, Middle/CK, and High/CK are displayed in Figure 4a–c, respectively. The metabolic pathways, such as arginine and proline metabolism, tricarboxylic acid (TCA) cycle, linoleic acid metabolism, and starch and sucrose metabolism, were enriched in all three comparisons. Moreover, each comparison had some unique metabolic pathways. For example, glycerophospholipid metabolism, tyrosine metabolism, histidine metabolism, and isoquinoline alkaloid biosynthesis were enriched in the comparison of Middle/CK. Amino sugar and nucleotide sugar metabolism, glycerolipid metabolism and galactose metabolism, tryptophan metabolism were enriched in the comparison of High/CK.

### 3.4. Correlation Analysis

The correlations between chlorpyrifos doses, metabolites with significant changes and physiological indices were determined using Pearson’s coefficient. Chlorpyrifos doses were significantly positively correlated with MDA (r = 0.916, *p* < 0.05), POD (r = 0.486, *p* < 0.05), and SOD (r = 0.988, *p* < 0.05) and significantly negatively correlated with chlorophyll (r = −0.703, *p* < 0.05), protein (r = −0.936, *p* < 0.05), rice height (r = −0.996, *p* < 0.05), and weight (r = −0.948, *p* < 0.05). In addition, 68 metabolites were significantly correlated with chlorpyrifos doses, where there were 36 and 32 metabolites for positive and negative relationships, respectively. In the positive correlations, these compounds primarily included quinic acid, phosphoenolpyruvic acid, proline, hydroquinidine, and 27 lipids (Figure 4). Compounds with negative correlations mainly included 8 flavonoids (homoorientin, isorhamnetin-3-O-glucoside, cyanidin 3-(2G-glucosylrutinoside), isoorientin, corymboside, biotin, hispidulin 4’-glucoside and peonidin-3-O-glucoside), 2 nucleosides (N2,N2-dimethylguanosine and choline), 4 amino acids (L-5-oxoproline and aspartate, glutamic acid and N,N-dimethylglycine), 3 phenolic acids, and 2 vitamins (Figure 5). In addition, many metabolites were significantly correlated with MDA, POD, SOD, protein, chlorophyll, rice height, and weight.

## 4. Discussion

As important agricultural inputs, pesticides play a vital role in protecting crop plants from diseases and pests; however, the effect of pesticides on crop plant physiology and metabolism remains undefined [18]. In the present study, the dose-effect of insecticide chlorpyrifos on rice plant physiology and metabolism was investigated. Our data revealed that chlorpyrifos could inhibit rice growth at high concentrations, in agreement with previous studies showing that three pesticides, emamectin benzoate, alpha-cypermethrin, and imidacloprid, could affect tomato growth [19]. The level of reactive oxygen species (ROS) was enhanced when plants were challenged by various stressors, and these reactive oxygen species could cause damage to cell components [20]. Plants in response to ROS burst could regulate antioxidant defense systems to produce antioxidant enzymes, such as SOD and POD, to eliminate the toxic effects of ROS [17]. In our data, the enzyme activities of both POD and SOD in rice tissues increased with increasing chlorpyrifos doses, indicating that chlorpyrifos could cause oxidative stress in rice plants. MDA as an end-product of membrane lipid peroxidation could reflect the current degree of lipid peroxidation [21]. The MDA content in rice tissues increased with increasing chlorpyrifos doses, indicating an increase in membrane lipid peroxidation. This result is in line with previous studies showing that high concentrations of polycyclic aromatic hydrocarbons led to significantly increased MDA production, which resulted in damage to cell membranes [22]. Both protein and chlorophyll play important roles in plant development [23]. The level of protein was decreased in middle and high groups, compared to the control, indicating that high concentrations of chlorpyrifos exerted a negative effect on protein synthesis. Chlorophyll levels increased at Low/CK and Middle/CK but decreased at High/CK. This suggests that photosynthesis was enhanced at low concentrations of chlorpyrifos and was inhibited at higher concentrations, which is termed hormesis [24]. Our result is in line with the research from Saenz et al. [25], where the photosynthesis of green algae was stimulated by 50% in response to a low concentration of the pesticide cyfluthrin.

PCA analysis displayed that the metabolic profiling of rice tissues was clearly altered under chlorpyrifos stress. The samples in middle and high groups were relatively close in distance, which indicates that the metabolic profiling of the two groups was similar. Amino acids play many important roles in plant growth and development [26,27]. In our data, several amino acids from the glutamate family (glutamate, proline, glutathione, and arginine) were significantly affected under chlorpyrifos stress. Both proline and glutathione are important antioxidants in plants, preventing cellular damage caused by ROS [28]. Glutathione was significantly increased in response to chlorpyrifos at three concentrations compared to the control. However, proline was only upregulated in Middle/CK and High/CK comparison. The accumulation of proline and glutathione may help to stabilize enzymes and membrane structures under chlorpyrifos stress [19,29]. Glutamic acid is involved in plant ammonium assimilation [30]. In addition, glutamic acid, a central amino acid, is converted to other amino acids through biochemical reactions [31]. Glutamic acid and its derivatives, such as N-acetyl-DL-glutamic acid and N-methyl-L-glutamic acid, were significantly decreased under chlorpyrifos stress at all three concentrations compared to the control. The reduction of glutamic acid and its derivatives may indicate a decreased capacity for nitrogen assimilation through the glutamate synthase pathway [32]. Arginine serves not only as a stored component of nitrogen in plant cells and recycling but also as a precursor of the biosynthesis of polyamines and nitric oxide [33]. Arginine was significantly increased under chlorpyrifos stress at all three concentrations compared to the control, which may enhance plant resistance to environmental stress [34]. Moreover, aromatic amino acids (such as tyrosine, tryptophan, and phenylalanine) are among the important precursors for synthesizing secondary metabolites such as flavonoids and phytohormones [35]. The content of tyrosine were significantly accumulated in Middle/CK, but the content of tryptophan were significantly accumulated in High/CK, suggesting that the pathway of secondary metabolite synthesis may be affected by chlorpyrifos.

Organic acids, the products of incomplete oxidation of photosynthetic assimilates, are central to cellular metabolism and play an important role in plant stress resistance [36]. There existed ten organic acids with significant increase or decrease in the CP group, compared to the control. Malic acid, citric acid, and succinic acid are involved in the TCA cycle, which provides the energy that enables various cellular functions [37]. However, these three organic acids exhibited differential trends. Citric acid was significantly increased at Low/CK and was decreased at Middle/CK. Malic acid was significantly increased in Middle/CK and High/CK comparison. Succinic acid and its derivative methylsuccinic acid were only decreased at Middle/CK group. These findings reveal that TCA cycle intermediates should be associated with oxidative stress [38]. Moreover, these TCA cycle intermediates are related to plant carbon metabolism [39]. For example, malic acid could participate in amino acid biosynthesis. Most amino acids were decreased in High/CK, which may lead to the accumulation of their precursor substance malic acid. Our result is in line with research from Che-Othman, et al. [40], where salt treatment affects TCA cycle activity in wheat leaves. In addition, both aminobenzoic acid and quinic acid were significantly increased in response to chlorpyrifos at all three concentrations compared to the control, and phosphoenolpyruvic acid was significantly accumulated in Middle/CK and High/CK groups. All three organic acids are related to plant defenses [41]. Aminobenzoic acid, a precursor of folic acid and coenzyme Q, is involved in various physiological processes from the regulation of disease resistance and stress tolerance in plants [42,43]. Quinic acid, one of the major intermediates in the biosynthetic pathway of most plant aromatic compounds, is considered an antioxidant that eliminates oxidative stress [44]. Phosphoenolpyruvic acid, a glycolysis metabolite with a high-energy phosphate group, exerts cytoprotective effects and has antioxidative potential [45].

Lipids as important components of biomembranes could protect cells and their organelles from various stressors [46,47]. Fatty acids are building blocks for the majority of cellular lipids. [48]. In our study, most fatty acids were significantly increased under chlorpyrifos stress, many of which were unsaturated fatty acids. The increase of unsaturated fatty acids especially for linolenic acid and linoleic acid, has been reported to be associated with both abiotic and biotic stresses [49]. Linolenic acid is metabolized in a stepwise fashion to jasmonic acid, which could regulate plant growth and resistance [50,51]. Tan, et al. reported that Pb could cause oxidative stress to rice plants and enhanced the increase in unsaturated fatty acids, which is consistent with our results [52]. In addition, many other lipids with significant increase were identified, mainly lysophospholipids (LPG, LPE, and LPC) and phospholipids (PC, PE, PG, and PI) [53]. Lysophospholipids are derived from glycerophospholipids through the action of phospholipase As [54]. Many lysophospholipids and glycerophospholipids with significant increase were derivatives of linolenic acid and linoleic acid. Both lysophospholipids and glycerophospholipids are involved in multiple regulatory processes, such as cell signaling, intracellular trafficking and scavenging of active oxygen [55]. Liu et al. reported that polycyclic aromatic hydrocarbons could cause an increase in phospholipids (PC and PE), which is consistent with our results [22]. Bactericide tricyclazole remarkably could affect fatty acid metabolism pathways [16]. Therefore, the above data reveal that the accumulation of fatty acids, lysophospholipids and glycerophospholipids could be beneficial for rice plants to relieve chlorpyrifos stress.

Sugars not only provide carbon and energy sources for plant metabolism but also act as signaling molecules for regulating plant growth [56]. In our findings, three sugars with significant changes were observed, including sucrose, melezitose, and D-glucose 6-phosphate. All of them were increased under chlorpyrifos stress, although both sucrose and melezitose were changed in the High/CK comparison. The increase of these three sugars may provide energy to resist the stress of chlorpyrifos. In addition, the accumulation of these sugars helped maintain the osmotic pressure balance of plant cells under pesticide stress [57]. Our result is in line with previous studies showing that both mancozeb and imidacloprid cause an increase in sugar content in plant shoots [58,59].

Flavonoids are among important secondary metabolites in plants and have many functions, such as regulating cell growth and attracting pollinator insects [60]. In our study, 23 flavonoids were found, and most flavonoids were downregulated in the comparison of Middle/CK and High/CK groups. This suggested that chlorpyrifos at high levels may inhibit the pathway of flavonoid synthesis in rice plants. Zhang et al. reported that the content of flavonoids in lettuce was significantly decreased under imidacloprid and fenvalerate stress, consistent with the results of our study [15].

## 5. Conclusions

Our results revealed that chlorpyrifos a low dose slightly could disrupt plant physiology and that chlorpyrifos a high dose could cause oxidative stress in rice plants and inhibits the synthesis of protein and chlorophyll. Many osmolality substances were significantly increased, including proline and glutathione, unsaturated fatty acids (linolenic acid and linoleic acid), and their derivatives lysophospholipids and glycerophospholipids, which could help rice cells relieve chlorpyrifos membrane damage. Nitrogen assimilation through the glutamate synthase pathway was significantly affected, and glutamic acid and its derivatives (N-acetyl-DL-glutamic acid and N-methyl-L-glutamic acid) were significantly decreased. Chlorpyrifos at moderate and high concentrations could cause the downregulation of flavonoids, which may be used to scavenge reactive oxygen species produced by chlorpyrifos stress. In addition, aminobenzoic acid, quinic acid, and phosphoenolpyruvic acid were significantly increased, which may act as biomarkers for rice plants stressed by chlorpyrifos. The conceptual model of chlorpyrifos effect on rice physiology and metabolic profiling was summarized in Figure 6. A limitation of this study is to explain the effect of chlorpyrifos on rice metabolism only using metabolomic data. The follow-up work will further analyze the regulation mechanism of pesticides on plant metabolism using multi-omics, such as transcriptomics, proteomics, and metabolomics.

## Figures and Tables

**Figure 1 metabolites-12-01289-f001:**
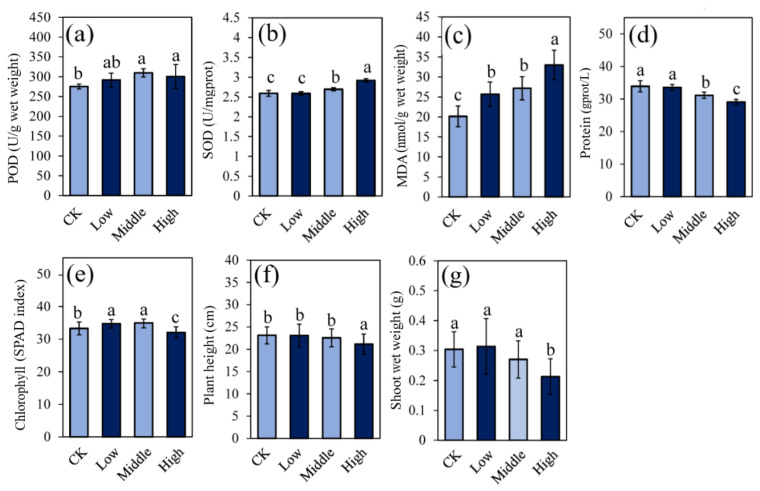
Effect of chlorpyrifos on plant growth and physiological activities, including POD (**a**), SOD (**b**), MDA (**c**), protein (**d**), chlorophyll (**e**), plant height (**f**), and shoot wet weight (**g**). Low, middle, and high indicate the chlorpyrifos groups at low, middle, and high concentrations, respectively, and CK indicates the control group. Bars with different letters are significantly different (*p* < 0.05), which was determined by one-way ANOVA with Tukey post hoc. Each treatment included with five independent biological replicates for POD, SOD, MDA, and protein analysis, and with 20 independent biological replicates for chlorophyll, plant height, and shoot weight analysis.

**Figure 2 metabolites-12-01289-f002:**
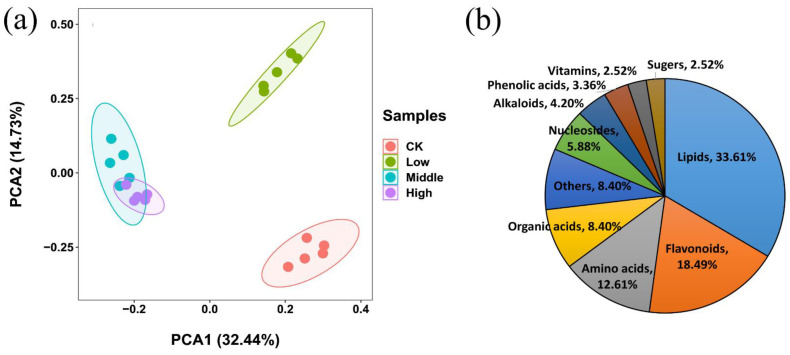
(**a**) PCA analysis of metabolic profiling of the rice shoots under chlorpyrifos stress: low, middle, and high indicate the chlorpyrifos groups at low, middle, and high concentrations, respectively, and CK indicates the control group. (**b**) Percentage of metabolites with significant changes belonging to different categories.

**Figure 3 metabolites-12-01289-f003:**
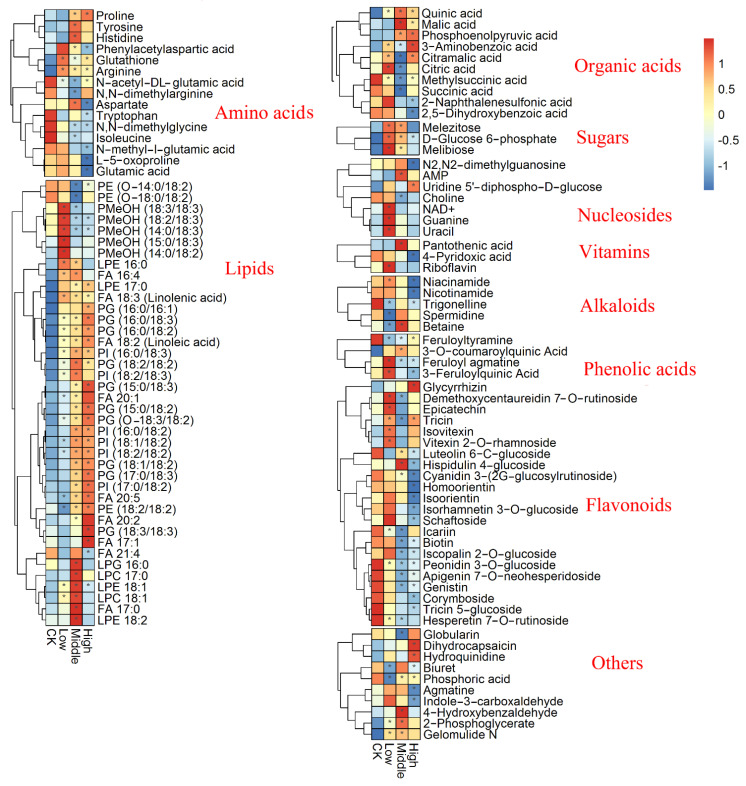
Heatmap analysis of the identified metabolites in rice shoots, where the subbox with * indicates metabolites with significant differences compared to the control. Low, middle, and high indicate the chlorpyrifos groups at low, middle, and high concentrations, respectively, and CK indicates the control group.

**Figure 4 metabolites-12-01289-f004:**
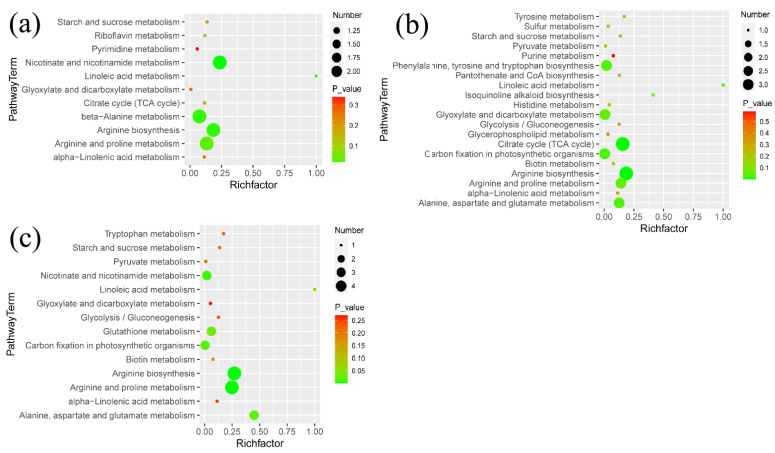
(**a**) Pathway analysis of the metabolites with significant differences in different comparisons: (**a**–**c**) are for the treatments at low, middle, and high exposure to chlorpyrifos, respectively, compared to the control, where Number is number of compound matched from our data and *p* values is *p* value calculated from pathway analysis.

**Figure 5 metabolites-12-01289-f005:**
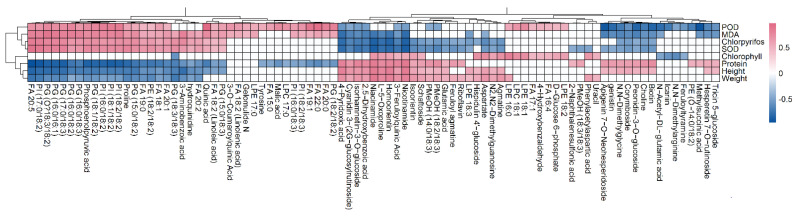
Correlation analysis of metabolites with rice physiological activities, plant height (Height), and shoot weight (Weight) via Pearson’s correlation coefficient (*p* < 0.05): the red and blue blocks indicate significant positive and negative correlations, respectively, and the white color indicates metabolites with no significant correlation.

**Figure 6 metabolites-12-01289-f006:**
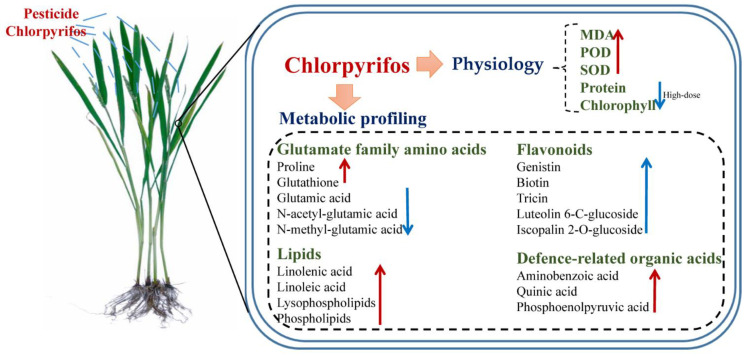
Conceptual model of chlorpyrifos effect on rice physiology and metabolic profiling, where red upward arrow and blue downward arrow denoted up and downregulation, respectively.

## Data Availability

The original contributions presented in the study are included in the article; further inquiries can be directed to the corresponding authors.

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
