# Peer review of "Metabolomic Analysis Reveals the Effect of Insecticide Chlorpyrifos on Rice Plant Metabolism"

_metabolites, 2022, doi:10.3390/metabo12121289_

Round 1
Reviewer 1 Report
Dear authors,
I have read the manuscript "Metabolomics analysis reveals the effect of insecticide chlorpyrifos on rice plant metabolism".
There are some suggestions that I consider will improve your manuscript.
Abstract
The authors must make an effort to highlight the novelty of their results versus current knowledge. Moreover, it should contain some data, concerning the results.
The keywords should not overlap with the title
Introduction
In the "Introduction" part, there is lack of information regarding the novelty of the experiment. A high-quality paper has to provide a proper state-of-the-art analysis after the literature review which should correspond to the paper goals. Furthermore, the authors laconically present information about chlorpyrifos and the need for its study. Moreover, at the end of the introduction, the main assumptions of the study and hypotheses have to be properly formulated.
As a general comment: In order to justify the need for this study multiple details are lacking The authors must provide information about chlorpyrifos (characteristics of insecticide, background of studies previous with this insecticide).
L34-36. Please, detail these effects
Materials and methods
L68-87: Detail the methodology to guarantee the reproducibility of the assay.
Results and discussion
The authors present a good discussion about the mechanisms of protection against oxidative stress within the plant. However, the authors do not compare how treatments could trigger or trigger these mechanisms.
Conclusion
Please re-check the conclusion. Moreover, the final remarks the authors should include details of any limitations of this study and recommendations for future perspectives.
Author Response
1) I have read the manuscript "Metabolomics analysis reveals the effect of insecticide chlorpyrifos on rice plant metabolism".
There are some suggestions that I consider will improve your manuscript.
Response: Thank you very much for your comments.
2) Abstract
The authors must make an effort to highlight the novelty of their results versus current knowledge. Moreover, it should contain some data, concerning the results.
Response: Thank you very much for your comments. The abstract has been revised and please see.
3) The keywords should not overlap with the title
Response: Thank you very much for your comments. The keyword has been revised and please see.
4) Introduction
In the "Introduction" part, there is lack of information regarding the novelty of the experiment. A high-quality paper has to provide a proper state-of-the-art analysis after the literature review which should correspond to the paper goals. Furthermore, the authors laconically present information about chlorpyrifos and the need for its study. Moreover, at the end of the introduction, the main assumptions of the study and hypotheses have to be properly formulated.
Response: Thank you very much for your comments. Besides, the part of Introduction has been revised as you suggestion and please see the part of Introduction.
4) As a general comment: In order to justify the need for this study multiple details are lacking The authors must provide information about chlorpyrifos (characteristics of insecticide, background of studies previous with this insecticide).
Response: Thank you very much for your comments. As you suggestion, more information about characteristics of insecticide, background of studies previous with this insecticide has been added. Please see Line 32-47.
5) L34-36. Please, detail these effects
Response: Thank you very much for your comments. this part has been revised. As you suggestion, more information about characteristics of insecticide, background of studies previous with this insecticide has been added. Please see Line 32-47.
6) Materials and methods. L68-87: Detail the methodology to guarantee the reproducibility of the assay.
Response: Thank you very much for your comments. The part of Materials and methods has been revised and please see Line 78-130.
7) Results and discussion. The authors present a good discussion about the mechanisms of protection against oxidative stress within the plant. However, the authors do not compare how treatments could trigger or trigger these mechanisms.
Response: Thank you very much for your comments. Interaction among the metabolites under stress were discussed and please see the part of Discussion (such as Line 275-280, 290-294 and 318-319).
8) Conclusion. Please re-check the conclusion. Moreover, the final remarks the authors should include details of any limitations of this study and recommendations for future perspectives.
Response: Thank you very much for your comments. The information of limitations of this study and recommendations for future perspectives has been added and please see the part of Conclusion.

Reviewer 2 Report
The manuscript is well written but I have some major and minor points to be addressed by the authors.
Major points:
Material and Methods are poorly described. It is missing important information about all sections, including compound identification, statistical analysis etc. This lack of information do not allow me to be confident about your results. But my main issue is about compound identification. It is necessary to add supporting information declaring the retention times, the m/z observed, the mass error in ppm or mDa, and the identified adduct for each compound identified. Which compounds did you corroborate their identity with standards? The unconfirmed compounds must be mentioned as tentatively identified. Also it is necessary to include the chemotaxonomic criterion in the compound identification process. Just an example. The case of genistin, which is an isoflavone, not a flavonoid. Isoflavones are polyphenolic compounds derived from the Fabaceae family. So how can you explain its presence in rice. Actually, there are several efforts by genetic transformation to produce isofavones in rice because these metabolites regulate the interaction among Leguminosae plants and beneficial bacteria. Please add supporting information of each compound identified including the chemotaxonomic criterion.
The resolution of all figures must be improved. There is a missing figure in Figure 1 and please review the statistical analysis because I am not sure about the differences you declare in the text of figure 1. The deviation bar is so big and makes difficult to believe in differences among treatments. In addition, you do not declare the post-hoc analysis neither if you test the data normality.
In figure 2 the compound names must be homogenized. Some are in uppercase, some in lowercase and again, it is needed supporting information of each comparison including the fold change values for each compound in each comparison.
Figure 3A to 3C are so small and the resolution should be improved. What was considered for rich factor? The number of metabolites identified in a metabolic pathway or the above plus the fold change value. Metaboanalyst can perform the analysis considering just the number of metabolites identified (Pathway analysis module) of can perform a deeper analysis considering the fold change values (Functional analysis). Which module did you use? Another important question is which data did you use? Did You use increased and decreased metabolites in each comparison? or you just considered over-accumulated metabolites in low/medium/high treatments? All these information is missing.
Because the lack of information about compound identification and statistical/informatic analysis I am not sure about the results, and in consequence in the discussion and conclusion information. Once you revised your results, I suggest adding a model that integrates your results.
Minor points:
All minor points are included in the attached file.

Author Response
1) The manuscript is well written but I have some major and minor points to be addressed by the authors.
Response: Thank you very much for your comments.
2) Major points:
Material and Methods are poorly described. It is missing important information about all sections, including compound identification, statistical analysis etc. This lack of information do not allow me to be confident about your results. But my main issue is about compound identification. It is necessary to add supporting information declaring the retention times, the m/z observed, the mass error in ppm or mDa, and the identified adduct for each compound identified. Which compounds did you corroborate their identity with standards? The unconfirmed compounds must be mentioned as tentatively identified. Also it is necessary to include the chemotaxonomic criterion in the compound identification process. Just an example. The case of genistin, which is an isoflavone, not a flavonoid. Isoflavones are polyphenolic compounds derived from the Fabaceae family. So how can you explain its presence in rice. Actually, there are several efforts by genetic transformation to produce isofavones in rice because these metabolites regulate the interaction among Leguminosae plants and beneficial bacteria. Please add supporting information of each compound identified including the chemotaxonomic criterion.
Response: Thank you very much for your comments. The result of compound identification and statistical analysis has been added in Support information Table S1.
3) The resolution of all figures must be improved. There is a missing figure in Figure 1 and please review the statistical analysis because I am not sure about the differences you declare in the text of figure 1. The deviation bar is so big and makes difficult to believe in differences among treatments. In addition, you do not declare the post-hoc analysis neither if you test the data normality.
Response: Thank you very much for your comments. These figures have been revised. The one-way ANOVA with Tukey post hoc was used in this part. The statistical differences for plant height and weight, and chlorophyll have been checked and the result of statistical analysis on plant weight (Fig.1 (g)) has been revised. Please see Fig. 1.
4) In figure 2 the compound names must be homogenized. Some are in uppercase, some in lowercase and again, it is needed supporting information of each comparison including the fold change values for each compound in each comparison.
Response: Thank you very much for your comments. Figure 2 has been revised. The result of compound identification and statistical analysis has been added in Support information Table S1.
5) Figure 3A to 3C are so small and the resolution should be improved. What was considered for rich factor? The number of metabolites identified in a metabolic pathway or the above plus the fold change value. Metaboanalyst can perform the analysis considering just the number of metabolites identified (Pathway analysis module) of can perform a deeper analysis considering the fold change values (Functional analysis). Which module did you use? Another important question is which data did you use? Did You use increased and decreased metabolites in each comparison? or you just considered over-accumulated metabolites in low/medium/high treatments? All these information is missing.
Response: Thank you very much for your comments. Fig. 3 has been revised. Metabolic pathways analysis was analyzed using a web-based tool MetaboAnalyst (Pathway analysis module) based on the metabolites with significant changes (including increase and decrease) in the chlorpyrifos group compared to the control and referring to the Kyoto Encyclopedia of Genes and Genomes database. In this figure, Number is number of compound matched from our data and P_values is p value calculated from pathway analysis.
6) Because the lack of information about compound identification and statistical/informatic analysis, I am not sure about the results, and in consequence in the discussion and conclusion information. Once you revised your results, I suggest adding a model that integrates your results.
Response: Thank you very much for your comments. The result of compound identification and statistical analysis have been added in Support information Table S1. The conceptual model of chlorpyrifos effect on rice physiology and metabolic profiling was summarized in Fig. 6.
7) Minor points:
All minor points are included in the attached file.
Response: Thank you very much for your comments. Language problem has been revised.
8) Line 22, Here you are talking about two doses that never were previously mentioned. In fact, there are three doses.
Response: Thank you very much for your comments. The abstract has been revised and please see Line 12.
8) Line 35 “affect body systems” of what kind of organisms?
Response: Thank you very much for your comments. It has been reported that chlorpyrifos could affect motor function of the human body and lead to the accumulation of toxins in the liver due to its toxicity. It has been revised and please see Line 42-43.
9) Line 45 gallic acid hydrate
Is this compound present in nature? or just gallic acid?
Response: Thank you very much for your comments. it is just gallic acid and it is commonly present in plants.
10) Line 71
How much volume did you sprayed to rice plants?
Response: Thank you very much for your comments. it was 50 mL and it has been added. Please see Line 81.
11) How much volume did you injecte for each sample? What ionization source did you use? ESI? APCI?
Response: Thank you very much for your comments. The volume of each sample injected was 2 μL. Ihe ionization source is electrospray DuoSpray ion source (ESI). Please see Line 104 and 113.
12) Line 105 “The MS peaks were identified by”. What did you identified? Peaks or compounds? For samples alignment, first ther is a peak identification step, then a compound identification.
Response: Thank you very much for your comments. It has been revised as “The compound was identified by”.
13) Line 133-134:Details regarding data acquisition and analysis were performed as described previously.
I think that the information declared for data acquisition and analysis is not enough and it should be mentioned in the manuscript or in supplementary information.
Response: Thank you very much for your comments. It has been revised and please see the part of 2.4. Metabolomic analysis and the result of compound identification and statistical analysis has been added in Support information Table S1.
14) Line 122-125, Please check the units of MDA, protein and chlorophyll contents.
Response: Thank you very much for your comments. The units of MDA, protein and chlorophyll contents have been revised and please see Fig. 1.
15) Line 129. High POD and SOD activity suggest high oxidative stress but this was not determined. Please be more precise.
Response: Thank you very much for your comments. It has been revised. Our results demonstrated that chlorpyrifos at low doses could slightly affect plant physiology but causes oxidative stress in rice plants at both middle- and high-doses and inhibited plant growth and the synthesis of protein and chlorophyll at high-doses. Please see Line 142-145.
17) Line 130. I am not sure about the statistical differences for plant height and weight. It is not mentioned what you display for deviation (standard deviation or standard error) but what I see, just considering the deviation bar is a slight decrease in height and weight but I am not sure about statistical differences.
Response: Thank you very much for your comments. The statistical differences for plant height and weight, and chlorophyll have been checked and the result of statistical analysis on plant weight (Fig.1 (g)) has been revised. Please see Fig. 1.
18) Line 148: a fold change greater than 2.0. There should be supporting information declaring the fold change values for each comparison. In Figure 3 it is not possible to observe the individual FC values.
Response: Thank you very much for your comments. The result of compound identification has been added in Support information Table S1 and the relative abundance of each component in each treatment was also shown.
19) Do PCA consider positive or negative ionization modes data? Which information did you use to generate PCA?
Response: Thank you very much for your comments. The matrix of MS peak area in positive or negative ionization modes was analyzed using principal component analysis (PCA). Please see Line 120-121.
20) Fig. 4 (a), (b) and (c). These figures are so small and the resolution should be improved. What is it considere for rich factor. The number of metabolites identified in a metabolic pathway or the above plus the fold change value. Metaboanallyst can perform the analysis considering just the number of metabolites identified (Pathway analysis module) of can perform a deeper analysis considerig the fold change analysis (Functional analysis). Which module did you use? Another important question is which data did you use? Did You use increased and decreased metabolites in each comparison? or you just considered overaccumulated metabolites in low/medium/high treatments?
Response: Thank you very much for your comments. The figure has been revised. Metabolic pathways analysis was analyzed using a web-based tool MetaboAnalyst (Pathway analysis module) based on the metabolites with significant changes in the chlorpyrifos group compared to the control and referring to the Kyoto Encyclopedia of Genes and Genomes database. In this figure, Number is number of compound matched from our data and P_values is p value calculated from pathway analysis. Please see Fig. 4 and Line 123-126.
20) Line 136: 3.4. Correlation analysis. Which software/packge did you use to perform correlation analysis?
Response: Thank you very much for your comments. Pearson correlation analysis was used for analyzing the relationship among metabolites, plant growth indexes, physiological activities and chlorpyrifos doses, which were computed using the corr() function in MATLAB. it has been added in Line 126-128.
21) Line 191-192. What type of weight. in figure 3g is show the shoots weight but in the figure caption also is declared a non-shown figure corresponding to the wet weight.
Response: Thank you very much for your comments. It is shoot weight. It has been revised. please see Fig. 5.
22) Line 218-220 This is not clear. You are talking about your data but there is a reference [19]. So?
Response: Thank you very much for your comments. The reference has been deleted.
23) Did you find jasmonic acid in the LC-MS study? If linoleic acid is accumulated according to chlorpyrifos concentration so why jasmonic acid was not detected?
Response: Thank you very much for your comments. Jasmonic acid was not detected in our data, which may be due that the content of jasmonic acid was very low or this component was without MS/MS (Only the component with MS/MS was identified).
24) Line 308-309: This decrease of these flavonoids may be used to scavenge reactive oxygen species produced by chlorpyrifos stress [57].
How is it possible? Your data suggest increasing oxidative stress according to chlorpyrifos concentration, and a flavonoids content decrease. So how a flavonoid decrease is a response to chlorpyrifos stress?
Response: Thank you very much for your comments. In fact, our data only showed that chlorpyrifos at high levels may inhibit the pathway of flavonoid synthesis in rice plants. This sentence has been deleted.

Round 2
Reviewer 1 Report
The revisions are satisfactory and the revised manuscript can be accepted to publish.
Reviewer 2 Report
Authors considered all my suggestions.